# Effects of Temperature and Radiation on Yield of Spring Wheat at Different Latitudes

Zhenzhen Zhang, Nianbing Zhou, Zhipeng Xing , Bingliang Liu, Jinyu Tian , Haiyan Wei, Hui Gao and Hongcheng Zhang *

Jiangsu Key Laboratory of Crop Cultivation and Physiology, Jiangsu Co-Innovation Center for Modern Production Technology of Grain Crops, Research Institute of Rice Industrial Engineering Technology, Yangzhou University, Yangzhou 225009, China; zz_zhang137@163.com (Z.Z.); 008007@yzu.edu.cn (N.Z.); zpxing@yzu.edu.cn (Z.X.); blliu@yzu.edu.cn (B.L.); dx120190081@yzu.edu.cn (J.T.); wei_haiyan@163.com (H.W.); gaohui@yzu.edu.cn (H.G.)
* Correspondence: hczhang@yzu.edu.cn; Tel.: +86-0514-87974595

**Abstract:** It is of great importance to investigate spring wheat yield affected by the climate at different latitudes in the Rice-Wheat Rotation System. Two spring wheat varieties used as the study objects were planted at two locations of different latitudes in 2017–2018 and 2018–2019. Six sowing dates were selected for planting the wheat seeds. The quantity of basic seedlings for the first sowing date was $300 \times 10^4$ ha$^{-1}$, which was increased by 10% on each date in the following sowing proceeding. Results showed that as the latitude increased, the mean daily temperature and effective accumulated temperature decreased, the mean solar radiation and accumulated solar radiation increased; as the effective accumulated temperature decreased, the yield decreased by 0.18 t ha$^{-1}$ on average; and dry matter accumulation decreased by 0.6 t ha$^{-1}$ on average. As the sowing date was delayed, the mean daily temperature and mean daily solar radiation increased, and the effective accumulated temperature and accumulated solar radiation decreased. Due to the decrease in the accumulated solar radiation and increase in mean daily temperature, the yield decreased by 0.27 t ha$^{-1}$ on average and the dry matter decreased by 0.39 t ha$^{-1}$ on average by postponing one sowing date. The effective accumulated temperature and accumulated solar radiation were significantly positively correlated with wheat yield and dry matter accumulation, and the mean daily temperature was significantly negatively correlated with wheat yield and dry matter accumulation. The temperature productivity at a high latitude was higher than lower latitude. The radiation productivity at a high latitude was lower than lower latitude. The productivity of the temperature and radiation first increased and then decreased when the sowing time was delayed.

**Keywords:** spring wheat yield; temperature; solar radiation

## 1. Introduction

The impact of climate change on food production has been a great concern for people since the recognition of global warming [1,2]. Gaffen DJ et al. predicted that the temperature would continue to rise based on the climate model and extreme weather would occur more frequently [3]. In the past 50 years, the annual average temperature has increased at a rate of 0.29 °C per decade. It is estimated that sunshine duration has decreased by 67.1 h per decade according to linear trend estimation, especially in the last 15 years [4]. The decrease in sunshine duration also means a decrease in solar radiation. In the situation where temperature rises and radiation decreases, it is of utmost importance to study the effect of temperature and radiation on the output of wheat.

The change in temperature and radiation is an influential factor in the growth and production of wheat. The study of Asseng shows that when the temperature increases by 2 °C in the growth season, the yield decreases by 50% [5,6]. The mean daily temperature suitable for wheat growth is 16–18 °C. Furthermore, 0 °C is the minimum temperature required for

wheat to sprout and wheat will stop sprouting and growing if the temperature is below 0 °C [7]. The optimum temperature for wheat to tiller is 13–18 °C, and tillering becomes suppressed if that temperature is exceeded. The temperature is suited to all stages of wheat growth if planted on the suitable date. If planted too late, the accumulated temperature will be insufficient, which will affect the wheat yield and dry matter accumulation [8,9]. In the reproductive growth stage of wheat, a high temperature affects the photosynthesis of wheat, which will cause the infertility of seeds and reduce the production of carbohydrates [10,11]. Radiation also affects the substance generation of wheat [12].

Jiangsu Province is one of the largest wheat production provinces in China, and the area of wheat growth accounts for about 8.6% of the total wheat growth area in China, ranking the fifth on the list [13]. Jiangsu Province takes the Huai River as the dividing line across the winter sowing spring wheat area and semi-winter wheat area. To the south of Huai River, there is a transition zone between the subtropical and warm temperate zone, mainly for spring wheat growth. To the north of Huai River, there is a warm temperate humid monsoon climate region, mainly used for planting semi-winter wheat varieties. Because the Huai River Basin is sensitive and vulnerable to climate change, changes in latitude will change the suitable planting area of different wheat types [14]. Since the 1990s, the development and application of mechanized high-yield rice cultivation techniques and the rice cultivation replacement project through which indica rice was changed to japonica rice in Jiangsu province led to rice harvest delays [15–17]. This led to a late planting of wheat in the Rice-Wheat Rotation System. The delay in the sowing date also caused a change in temperature and radiation during the growth period of wheat, which affected the yield of wheat. Although there have been a lot of studies on the effects of temperature and radiation on wheat yield, there are few studies on the effects of temperature and radiation on wheat yield under the conditions of relatively high yield cultivation, super late sowing and spring wheat moving northward in this region. Therefore, the study set up different planting dates to sow the spring wheat at different latitudes, aiming to explain the following: (1) differences of temperature and radiation at different sowing dates and at different latitudes, (2) the effects of temperature and radiation on the yield of spring wheat, and (3) the feasibility of northward migration of spring wheat from the perspective of temperature and radiation.

## 2. Materials and Methods

### 2.1. Experimental Site and Experimental Design

Experiments were carried out in Jianhu County, Yancheng City, Jiangsu Province (33°47′ N, 119°77′ E) and Donghai County, Lianyungang City, Jiangsu Province (34°35′ N, 118°45′ E) from 2017 to 2018 and 2018 to 2019. Both differ in latitude. Donghai County (HL) is located at a high latitude, has a warm temperate, humid monsoon climate and clayey soil. The field soil was a clay loam with nitrogen (1.55 g kg$^{-1}$), potassium (90.6 mg kg$^{-1}$), phosphorus (48.6 mg kg$^{-1}$), and organic matter (23.8 g kg$^{-1}$) in the topsoil (0–20 cm). The yield of rice in the previous crop in the experimental site was 9.95 t·ha$^{-1}$. The experimental site is mainly planted with semi-winter wheat. Jianhu County (LL) is situated in the transitional zone between subtropics and the warm temperate zone. The field soil was a clay loam with total nitrogen (1.59 g kg$^{-1}$), rapidly available potassium (96.6 mg kg$^{-1}$), rapidly available phosphorus (45.6 mg kg$^{-1}$), and organic matter (26.8 g kg$^{-1}$) found in the topsoil (0–20 cm). The yield of rice in the previous crop in the experimental site was 10.26 t ha$^{-1}$. Spring varieties and semi-winter varieties can grow in the experimental site, but spring varieties are given priority. Two spring varieties were selected for the experiment, namely Yangmai 23 (YM23) and Yangmai 25 (YM25), both of which have been widely planted in this area. Six sowing dates were selected for planting the wheat seeds, which were 31 October (S1), 10 November (S2), 20 November (S3), 30 November (S4), 10 December (S5), and 20 December (S6). The quantity of basic seedlings for the first sowing date was $300 \times 10^4$ ha$^{-1}$, and increased by 10% for each date in the following sowing proceeding. During this period, seedlings were manually thinned. The slow and

controlled release of fertilizer for wheat, made up of (N:P:K = 26:15:8) 750 kg·ha$^{-1}$ and 75 kg·ha$^{-1}$ urea (229.5 N ha$^{-1}$ in total), were applied to the experimental field as base fertilizer; but no fertilizer was applied in the later stages. The seeds were planted in the row spacing of 22 cm and the sowing depth of about 2 cm. The plot area was 15 m$^2$, and each variety was repeated twice.

### 2.2. Experimental Data Collection

The key growth periods of wheat, such as sowing date (SO), emergence stage (EM), jointing stage (JO), heading stage (HE), and mature stage (MA), were recorded. At different growth stages, 20 representative single stems were taken from each plot, they were killed for half an hour at 105 °C and then dried to constant weight at 80 °C. The dry matter weight at different stages was calculated according to the number of tillers at different stages. At the mature stage, the distance of one meter in each plot was taken to count the number of spikes, which was repeated four times to calculate the number of panicles per hectare according to row spacing. A total of fifty panicles were selected continuously in each plot to determine the number of grains per spike which was repeated three times. A total of four squares were taken from each plot for harvesting. The actual yield was determined after the grain moisture was dried to 14%.

The data of mean daily temperature and sunshine hours of wheat growing season in the two ecological points were provided by the meteorological observation stations in Donghai County, Lianyungang City, Jiangsu Province, China and Jianhu County, Yancheng City, Jiangsu Province, China. Based on the analysis of local meteorological data in the past 10 years, the temperature and radiation in this experiment were within the range of variation. Since radiation cannot be recorded directly, the Angstrom–Prescott (AP) equation was used to calculate the total daily radiation by latitude and sunshine hours.

### 2.3. Calculation Methods and Statistical Analysis

In this experiment, the Angstrom-Prescott (AP) equation was used to calculate solar radiation:

$$Q = QA(a + bs)$$

where Q (MJ m$^{-2}$d$^{-1}$) is the total solar radiation reaching the surface; QA (MJ m$^{-2}$d$^{-1}$) is astronomical radiation; "s" is the percentage of sunshine (%), "a" and "b" are empirical coefficients [18].

The formula for calculating the cumulative solar radiation (CSR) and mean daily solar radiation at each growth stage is as follows:

$$CSR = \sum Q \times \text{days of different growth stage (d)}$$

$$R_{mean} = \sum Q / \text{days of different growth stage (d)}$$

The formula for calculating the effective accumulative temperature and mean daily temperature in each growth stage is as follows:

$$EAT = \sum (T1 - T0) \times \text{days of growth stage (d) } [T1 > T0, \text{ when } T1 < T0, (T1 - T0) \text{ is calculated as 0].}$$

$$T_{mean} = \sum T1 / \text{days of different growth stage (d)}$$

where T1 and T0 (the lower limit temperature of wheat growth is 0 °C) are the mean daily temperature and biological zero temperature, respectively [19].

Negative accumulated temperature refers to the absolute value of the sum of mean daily temperatures of < 0 °C in a year. The calculation formula is

$$\text{Negative accumulated temperature} = \sum (T1 - T0) \times \text{Growth period } [T1 < T0, \text{ when } T1 > T0, (T1 - T0) \text{ is calculated as 0]}$$

$$\text{Utilization efficiency of temperature} = \text{Yield} / EAT \times 100\%$$

$$\text{Utilization efficiency of radiation} = \text{Yield} / CSR \times 100\%$$

The data were recorded and sorted out using Microsoft Excel 2016 and statistically analyzed by SPSS22.0 (ANOVA). The means were compared by the least significant difference at the probability level of 0.05 (LSD, $p = 0.05$). Sigma Plot 10.0 was used for graphing.

## 3. Results

### 3.1. Differences of EAT and $T_{mean}$ at Different Latitudes and Sowing Dates

The EAT of the two spring wheat cultivars showed the same pattern under different situations of latitudes, sowing dates and years (Table 1). When planted on the same date, the EAT in the whole growth stage was higher at HL than LL, and the maximum effective accumulative temperature was 101 °C d. At the same latitude, the EAT in the whole growth stage gradually decreased with the postponement of the sowing date. Compared with S1, the EAT of S2–S6 decreased by 64–110 °C d, 137–199 °C d, 172–267 °C d, 178–342 °C d and 168–354 °C d, respectively. The EAT from EM to JO, JO to HE and HE to MA decreased gradually.

**Table 1.** Difference in the effective accumulative temperature at different latitudes and sowing dates (°C d).

| Varieties | Latitude | Treatment | 2017–2018 | | | | | 2018–2019 | | | | |
|---|---|---|---|---|---|---|---|---|---|---|---|---|
| | | | SO-EM | EM-JO | JO-HE | HE-MA | Whole Growth Period | SO-EM | EM-JO | JO-HE | HE-MA | Whole Growth Period |
| YM23 | HL | S1 | 141 | 495 | 480 | 830 | 1970 | 122 | 564 | 498 | 811 | 1995 |
| | | S2 | 147 | 476 | 415 | 804 | 1868 | 133 | 505 | 458 | 835 | 1931 |
| | | S3 | 163 | 461 | 384 | 795 | 1803 | 146 | 469 | 453 | 790 | 1858 |
| | | S4 | 175 | 438 | 375 | 783 | 1743 | 151 | 461 | 409 | 774 | 1796 |
| | | S5 | 197 | 428 | 369 | 777 | 1743 | 178 | 403 | 399 | 778 | 1758 |
| | | S6 | 224 | 413 | 366 | 779 | 1732 | 199 | 389 | 410 | 768 | 1766 |
| | LL | S1 | 119 | 610 | 449 | 870 | 2048 | 114 | 636 | 461 | 888 | 2098 |
| | | S2 | 124 | 541 | 411 | 862 | 1938 | 119 | 578 | 460 | 832 | 1989 |
| | | S3 | 133 | 487 | 382 | 846 | 1849 | 127 | 544 | 412 | 822 | 1905 |
| | | S4 | 142 | 435 | 399 | 813 | 1789 | 142 | 509 | 381 | 799 | 1831 |
| | | S5 | 154 | 443 | 365 | 807 | 1770 | 152 | 453 | 384 | 789 | 1779 |
| | | S6 | 170 | 441 | 361 | 794 | 1766 | 167 | 430 | 384 | 790 | 1771 |
| YM25 | HL | S1 | 141 | 514 | 477 | 839 | 1970 | 122 | 594 | 491 | 835 | 2041 |
| | | S2 | 147 | 502 | 409 | 810 | 1868 | 133 | 547 | 444 | 829 | 1953 |
| | | S3 | 163 | 471 | 397 | 801 | 1832 | 146 | 506 | 437 | 794 | 1882 |
| | | S4 | 175 | 459 | 370 | 794 | 1798 | 151 | 493 | 399 | 778 | 1821 |
| | | S5 | 197 | 440 | 377 | 779 | 1792 | 178 | 425 | 410 | 768 | 1781 |
| | | S6 | 224 | 424 | 374 | 780 | 1802 | 199 | 434 | 402 | 755 | 1790 |
| | LL | S1 | 119 | 640 | 433 | 879 | 2071 | 114 | 673 | 452 | 910 | 2148 |
| | | S2 | 124 | 561 | 410 | 867 | 1962 | 119 | 604 | 452 | 867 | 2042 |
| | | S3 | 133 | 526 | 399 | 837 | 1895 | 127 | 567 | 422 | 843 | 1960 |
| | | S4 | 142 | 480 | 369 | 823 | 1813 | 142 | 530 | 385 | 830 | 1887 |
| | | S5 | 154 | 494 | 332 | 813 | 1794 | 152 | 480 | 371 | 803 | 1807 |
| | | S6 | 170 | 457 | 364 | 799 | 1790 | 167 | 459 | 370 | 798 | 1794 |

The difference in $T_{mean}$ between the two cultivars was not that significant at different latitudes and sowing dates (Table 2). Planted on the same sowing date, the $T_{mean}$ at HL was 0.4–0.87 °C lower than LL. With the delayed sowing date at the same latitude, the change in $T_{mean}$ of S1 and S2 was not very obvious while the $T_{mean}$ of S3–S6 increased gradually. With the postponement of the sowing date, the $T_{mean}$ from SO to EM gradually decreased, and decreased first and increased from EM to JO.

### 3.2. Difference of CSR and $R_{mean}$ at Different Latitudes and Sowing Dates

The CSR of the two spring wheat varieties presented the same pattern at different latitudes and sowing date and the CSR in the whole growth stage had the same pattern as well (Table 3). During the whole growth period, the CSR at HL was higher than LL by 269–440 MJm$^{-2}$. The CSR from JO to HE and HE to MA at HL was higher than LL. The CSR at the same latitude in the whole growth stage gradually decreased as the sowing date was delayed. Compared with S1, the CSR of S2–S6 decreased by 56–116 MJm$^{-2}$, 102–169 MJm$^{-2}$, 156–267 MJm$^{-2}$, 166–331 MJm$^{-2}$ and 214–411 MJm$^{-2}$, respectively.

**Table 2.** Difference in mean daily temperature at different latitudes and sowing dates (°C).

| Varieties | Latitude | Treatment | 2017–2018 | | | | | 2018–2019 | | | | |
|---|---|---|---|---|---|---|---|---|---|---|---|---|
| | | | SO-EM | EM-JO | JO-HE | HE-MA | Whole Growth Period | SO-EM | EM-JO | JO-HE | HE-MA | Whole Growth Period |
| YM23 | HL | S1 | 14.10 | 3.21 | 14.54 | 20.24 | 8.81 | 12.16 | 3.90 | 13.09 | 20.80 | 8.95 |
| | | S2 | 8.18 | 3.32 | 14.81 | 20.61 | 8.70 | 9.48 | 3.67 | 13.48 | 21.41 | 8.99 |
| | | S3 | 2.39 | 5.26 | 16.00 | 20.93 | 8.73 | 9.71 | 3.51 | 14.16 | 21.95 | 9.03 |
| | | S4 | 1.13 | 10.43 | 17.05 | 21.16 | 8.82 | 1.55 | 7.18 | 14.62 | 22.10 | 9.13 |
| | | S5 | 1.45 | 11.25 | 16.79 | 21.57 | 9.27 | 1.50 | 11.19 | 15.33 | 22.22 | 9.42 |
| | | S6 | 1.87 | 12.52 | 17.40 | 21.63 | 9.71 | 1.91 | 11.80 | 15.75 | 22.59 | 9.99 |
| | LL | S1 | 14.84 | 4.53 | 13.21 | 19.77 | 9.49 | 16.26 | 5.11 | 12.45 | 18.89 | 9.81 |
| | | S2 | 11.27 | 4.13 | 14.68 | 20.05 | 9.37 | 11.93 | 4.83 | 13.51 | 19.35 | 9.71 |
| | | S3 | 6.34 | 4.21 | 15.28 | 20.15 | 9.35 | 11.56 | 4.70 | 14.21 | 19.56 | 9.73 |
| | | S4 | 4.26 | 4.61 | 15.34 | 20.33 | 9.48 | 6.38 | 5.08 | 14.65 | 19.98 | 9.80 |
| | | S5 | 1.71 | 9.24 | 15.21 | 20.70 | 9.86 | 2.87 | 7.04 | 14.78 | 20.77 | 10.01 |
| | | S6 | 2.01 | 10.75 | 15.02 | 20.90 | 10.37 | 2.75 | 8.77 | 14.78 | 21.35 | 10.51 |
| YM25 | HL | S1 | 14.10 | 3.30 | 14.90 | 20.45 | 8.77 | 12.16 | 4.04 | 13.26 | 21.41 | 9.08 |
| | | S2 | 8.18 | 3.47 | 15.14 | 20.77 | 8.66 | 9.48 | 3.92 | 13.45 | 21.83 | 9.05 |
| | | S3 | 2.39 | 5.31 | 16.55 | 21.08 | 8.83 | 9.71 | 3.74 | 14.55 | 22.07 | 9.11 |
| | | S4 | 1.13 | 10.68 | 16.80 | 21.47 | 9.07 | 1.55 | 7.33 | 15.33 | 22.22 | 9.21 |
| | | S5 | 1.45 | 11.27 | 17.15 | 21.63 | 9.49 | 1.50 | 11.19 | 15.75 | 22.59 | 9.50 |
| | | S6 | 1.87 | 12.47 | 17.83 | 21.66 | 10.06 | 1.91 | 12.05 | 16.08 | 22.88 | 10.07 |
| | LL | S1 | 14.84 | 4.69 | 13.12 | 19.98 | 9.56 | 16.26 | 5.24 | 12.91 | 19.36 | 9.96 |
| | | S2 | 11.27 | 4.23 | 15.20 | 20.15 | 9.44 | 11.93 | 4.97 | 13.68 | 19.70 | 9.87 |
| | | S3 | 6.34 | 4.49 | 15.34 | 20.42 | 9.49 | 11.56 | 4.82 | 14.54 | 20.08 | 9.91 |
| | | S4 | 4.26 | 4.96 | 15.35 | 20.58 | 9.56 | 6.38 | 5.20 | 14.79 | 20.74 | 9.99 |
| | | S5 | 1.71 | 9.68 | 15.11 | 20.85 | 9.94 | 2.87 | 7.24 | 14.84 | 21.14 | 10.11 |
| | | S6 | 2.01 | 10.63 | 15.83 | 21.02 | 10.46 | 2.75 | 9.01 | 14.81 | 21.56 | 10.59 |

**Table 3.** Differences in cumulative solar radiation at different latitudes and sowing dates (MJ m$^{-2}$).

| Varieties | Latitude | Treatment | 2017–2018 | | | | | 2018–2019 | | | | |
|---|---|---|---|---|---|---|---|---|---|---|---|---|
| | | | SO-EM | EM-JO | JO-HE | HE-MA | Whole Growth Period | SO-EM | EM-JO | JO-HE | HE-MA | Whole Growth Period |
| YM23 | HL | S1 | 148 | 1176 | 568 | 681 | 2578 | 92 | 1122 | 595 | 720 | 2528 |
| | | S2 | 176 | 1157 | 483 | 622 | 2462 | 122 | 1105 | 528 | 716 | 2472 |
| | | S3 | 487 | 868 | 439 | 614 | 2409 | 92 | 1163 | 503 | 668 | 2426 |
| | | S4 | 740 | 611 | 375 | 612 | 2310 | 470 | 837 | 419 | 643 | 2369 |
| | | S5 | 697 | 603 | 384 | 586 | 2247 | 679 | 614 | 404 | 643 | 2341 |
| | | S6 | 723 | 506 | 402 | 568 | 2167 | 676 | 575 | 427 | 621 | 2300 |
| | LL | S1 | 115 | 1042 | 459 | 626 | 2243 | 67 | 810 | 555 | 670 | 2101 |
| | | S2 | 71 | 1007 | 458 | 596 | 2131 | 82 | 834 | 525 | 591 | 2032 |
| | | S3 | 190 | 910 | 416 | 583 | 2100 | 84 | 854 | 439 | 610 | 1988 |
| | | S4 | 272 | 791 | 400 | 570 | 2033 | 119 | 844 | 353 | 623 | 1939 |
| | | S5 | 498 | 540 | 375 | 555 | 1967 | 351 | 647 | 330 | 607 | 1935 |
| | | S6 | 479 | 509 | 372 | 537 | 1898 | 356 | 603 | 308 | 621 | 1888 |
| YM25 | HL | S1 | 148 | 1213 | 553 | 663 | 2578 | 92 | 1153 | 590 | 716 | 2551 |
| | | S2 | 176 | 1189 | 474 | 623 | 2462 | 122 | 1159 | 498 | 721 | 2501 |
| | | S3 | 487 | 889 | 438 | 622 | 2436 | 92 | 1221 | 472 | 657 | 2442 |
| | | S4 | 740 | 629 | 378 | 614 | 2361 | 470 | 870 | 404 | 643 | 2387 |
| | | S5 | 697 | 607 | 406 | 568 | 2279 | 679 | 642 | 427 | 621 | 2369 |
| | | S6 | 723 | 519 | 394 | 575 | 2210 | 676 | 627 | 405 | 621 | 2329 |
| | LL | S1 | 115 | 1052 | 474 | 620 | 2262 | 67 | 857 | 535 | 686 | 2146 |
| | | S2 | 71 | 1036 | 444 | 607 | 2158 | 82 | 864 | 499 | 642 | 2087 |
| | | S3 | 190 | 952 | 400 | 596 | 2138 | 84 | 894 | 409 | 655 | 2042 |
| | | S4 | 272 | 837 | 371 | 578 | 2059 | 119 | 887 | 329 | 654 | 1990 |
| | | S5 | 498 | 578 | 343 | 561 | 1980 | 351 | 680 | 303 | 626 | 1959 |
| | | S6 | 479 | 518 | 389 | 524 | 1910 | 356 | 632 | 292 | 613 | 1893 |

The $R_{mean}$ in the whole growth stage at HL was 1.18–2.19 MJm$^{-2}$d$^{-1}$ higher than LL (Table 4). The $R_{mean}$ of each growth period from seedling emergence to maturity at HL was higher than LL. At the same latitude and the $R_{mean}$ increased gradually with the postponement of sowing date. Compared with S1, the $R_{mean}$ of S2–S6 increased by −0.08–0.27 MJm$^{-2}$d$^{-1}$, 0.18–0.49MJm$^{-2}$d$^{-1}$, 0.23–0.77MJm$^{-2}$d$^{-1}$, 0.49–1.32MJm$^{-2}$d$^{-1}$ and 0.68–1.76 MJm$^{-2}$d$^{-1}$, respectively.

**Table 4.** Differences in mean daily cumulative solar radiation at different latitudes and sowing dates (MJ m$^{-2}$ d$^{-1}$).

| Varieties | Latitude | Treatment | 2017-2018 | | | | | 2018-2019 | | | | |
|---|---|---|---|---|---|---|---|---|---|---|---|---|
| | | | SO-EM | EM-JO | JO-HE | HE-MA | Whole Growth Period | SO-EM | EM-JO | JO-HE | HE-MA | Whole Growth Period |
| YM23 | HL | S1 | 14.81 | 8.97 | 17.21 | 16.61 | 11.99 | 9.16 | 8.70 | 15.65 | 18.46 | 11.70 |
| | | S2 | 9.79 | 9.56 | 17.27 | 15.95 | 11.95 | 8.74 | 9.13 | 15.54 | 18.37 | 11.89 |
| | | S3 | 8.40 | 11.13 | 18.29 | 16.17 | 12.17 | 6.13 | 10.03 | 15.71 | 18.55 | 12.19 |
| | | S4 | 8.40 | 14.56 | 17.03 | 16.54 | 12.22 | 7.34 | 13.28 | 14.97 | 18.38 | 12.47 |
| | | S5 | 8.30 | 15.87 | 17.46 | 16.28 | 12.48 | 8.08 | 17.06 | 15.54 | 18.39 | 12.93 |
| | | S6 | 8.93 | 15.33 | 19.13 | 15.78 | 12.67 | 8.56 | 17.44 | 16.43 | 18.27 | 13.37 |
| | LL | S1 | 14.41 | 8.34 | 13.51 | 14.22 | 10.63 | 9.56 | 6.69 | 15.00 | 14.26 | 9.91 |
| | | S2 | 6.42 | 8.39 | 16.36 | 13.86 | 10.55 | 8.20 | 7.19 | 15.45 | 13.75 | 10.01 |
| | | S3 | 9.07 | 8.67 | 16.64 | 13.89 | 10.88 | 7.67 | 7.63 | 15.15 | 14.53 | 10.25 |
| | | S4 | 8.25 | 9.31 | 15.37 | 14.24 | 11.05 | 5.40 | 8.70 | 13.59 | 15.58 | 10.48 |
| | | S5 | 7.78 | 11.25 | 15.61 | 14.22 | 11.24 | 7.30 | 10.12 | 12.70 | 15.97 | 10.99 |
| | | S6 | 7.61 | 12.42 | 15.52 | 14.12 | 11.43 | 6.47 | 12.31 | 11.83 | 16.79 | 11.30 |
| YM25 | HL | S1 | 14.81 | 9.12 | 17.29 | 16.18 | 11.93 | 9.16 | 8.73 | 15.94 | 18.37 | 11.70 |
| | | S2 | 9.79 | 9.67 | 17.54 | 15.98 | 11.89 | 8.74 | 9.35 | 15.09 | 18.98 | 11.97 |
| | | S3 | 8.40 | 11.26 | 18.25 | 16.36 | 12.24 | 6.13 | 10.26 | 15.73 | 18.24 | 12.21 |
| | | S4 | 8.40 | 14.64 | 17.19 | 16.58 | 12.42 | 7.34 | 13.17 | 15.54 | 18.39 | 12.50 |
| | | S5 | 8.30 | 15.58 | 18.46 | 15.78 | 12.59 | 8.08 | 16.89 | 16.43 | 18.27 | 13.02 |
| | | S6 | 8.93 | 15.26 | 18.75 | 15.97 | 12.85 | 8.56 | 17.41 | 16.21 | 18.81 | 13.46 |
| | LL | S1 | 14.41 | 8.28 | 14.36 | 14.10 | 10.67 | 9.56 | 6.86 | 15.28 | 14.60 | 10.03 |
| | | S2 | 6.42 | 8.49 | 16.44 | 14.11 | 10.63 | 8.20 | 7.32 | 15.13 | 14.59 | 10.18 |
| | | S3 | 9.07 | 8.90 | 15.37 | 14.53 | 10.96 | 7.67 | 7.84 | 14.11 | 15.59 | 10.42 |
| | | S4 | 8.25 | 9.51 | 15.47 | 14.46 | 11.13 | 5.40 | 8.96 | 12.67 | 16.36 | 10.64 |
| | | S5 | 7.78 | 11.33 | 15.60 | 14.40 | 11.25 | 7.30 | 10.30 | 12.11 | 16.47 | 11.07 |
| | | S6 | 7.61 | 12.05 | 16.89 | 13.80 | 11.44 | 6.47 | 12.39 | 11.67 | 16.58 | 11.27 |

### 3.3. Differences in Yield under Different Temperature and Radiation Conditions

The yield of the two spring varieties showed the same pattern under different latitudes and sowing date conditions, and the annual change pattern was also the same (Tables 5 and 6). Planted on the same sowing date, the yield at HL was 0.06–0.41 t·ha$^{-1}$ lower than LL, the number of panicles at HL was $-3.48$–22.63 × 10$^4$ ha$^{-1}$ lower than LL, and the 1000-grain weight at HL was 0.32–1.99 g lower than LL. At the same latitude, the yield of spring wheat decreased gradually with the postponement of sowing time, and compared with S1, the yield of S2–S6 decreased by 0.22–0.31 t·ha$^{-1}$, 0.49–0.62 t·ha$^{-1}$, 0.78–0.99 t·ha$^{-1}$, 1.07–1.23 t·ha$^{-1}$ and 1.23–1.39 t·ha$^{-1}$, respectively. The grain number had the trend of increasing but this was not significant when the sowing date was delayed. As the sowing date was pushed back, the grain number per spike decreased gradually, and that of S2–S6 was 0.54–5.21 lower than that of S1. As the sowing date was pushed backward, the 1000-grain weight of S2–S6 was 0.34–4.72 g lower than that of S1.

**Table 5.** Effects of different temperature and radiation on wheat yield.

| Variety | Latitude | Treatment | 2017–2018 | | | | 2018–2019 | | | |
|---|---|---|---|---|---|---|---|---|---|---|
| | | | Panicles (× 10$^4$ ha$^{-1}$) | Grain per Panicle | 1000-Grain Weight (g) | Yield (t·ha$^{-1}$) | Panicles (× 10$^4$ ha$^{-1}$) | Grain per Panicle | 1000-Grain Weight (g) | Yield (t·ha$^{-1}$) |
| YM23 | HL | S1 | 482.98 cd | 37.62 a | 41.62 ab | 7.40 b | 480.38 c | 37.72 a | 42.17 bc | 7.51 b |
| | | S2 | 477.35 cd | 37.27 a | 41.28 b | 7.17 c | 477.31 c | 36.89 bc | 41.69 bcd | 7.29 c |
| | | S3 | 484.26 bcd | 35.95 b | 40.30 c | 6.85 e | 484.88 bc | 35.34 d | 40.94 de | 6.89 d |
| | | S4 | 485.24 bcd | 35.17 c | 39.17 d | 6.57 fg | 490.88 abc | 34.88 de | 39.53 gh | 6.62 e |
| | | S5 | 493.83 abc | 34.50 d | 38.42 e | 6.24 h | 486.38 bc | 33.96 fg | 39.28 gh | 6.38 fg |
| | | S6 | 497.07 ab | 33.70 e | 37.75 f | 6.17 h | 505.75 ab | 33.60 g | 38.44 i | 6.19 h |
| | LL | S1 | 485.50 bcd | 37.67 a | 42.10 a | 7.56 a | 486.63 bc | 37.30 ab | 43.70 a | 7.69 a |
| | | S2 | 490.52 abc | 36.33 b | 41.64 ab | 7.32 b | 491.58 abc | 36.55 c | 42.46 b | 7.47 b |
| | | S3 | 490.23 abc | 35.75 bc | 41.24 b | 7.00 d | 498.70 abc | 35.10 de | 41.51 cd | 7.19 c |
| | | S4 | 495.05 abc | 34.40 d | 40.56 c | 6.68 f | 505.10 ab | 34.60 ef | 40.42 ef | 6.91 d |
| | | S5 | 500.20 a | 33.40 ef | 39.43 d | 6.47 g | 509.00 a | 33.35 g | 39.99 fg | 6.49 ef |
| | | S6 | 502.90 a | 32.87 f | 38.97 de | 6.23 h | 509.83 a | 32.35 h | 38.98 hi | 6.32 gh |

**Table 5.** *Cont.*

| Variety | Latitude | Treatment | 2017–2018 | | | | 2018–2019 | | | |
|---|---|---|---|---|---|---|---|---|---|---|
| | | | Panicles (× 10⁴ ha⁻¹) | Grain per Panicle | 1000-Grain Weight (g) | Yield (t·ha⁻¹) | Panicles (× 10⁴ ha⁻¹) | Grain per Panicle | 1000-Grain Weight (g) | Yield (t·ha⁻¹) |
| YM25 | HL | S1 | 480.91 c | 37.72 a | 41.90 a | 7.46 b | 488.51 cd | 37.55 a | 41.99 b | 7.57 ab |
| | | S2 | 486.89 bc | 36.62 b | 41.06 b | 7.17 d | 493.24 bc | 36.44 b | 40.85 cd | 7.32 cd |
| | | S3 | 489.13 abc | 35.70 c | 40.63 bc | 6.79 e | 497.93 bc | 35.45 c | 40.49 cde | 6.81 e |
| | | S4 | 500.50 ab | 34.47 d | 39.33 de | 6.59 fg | 493.95 bc | 34.21 d | 40.03 def | 6.58 f |
| | | S5 | 507.10 a | 33.45 e | 38.54 fg | 6.32 h | 512.43 ab | 32.79 e | 39.19 f | 6.33 g |
| | | S6 | 508.02 a | 32.97 e | 37.73 g | 6.16 i | 504.24 abc | 32.44 e | 39.16 f | 6.20 g |
| | LL | S1 | 486.65 bc | 37.77 a | 42.26 a | 7.61 a | 486.30 d | 37.35 a | 43.76 a | 7.72 a |
| | | S2 | 487.15 bc | 36.60 b | 41.89 a | 7.34 c | 493.55 bc | 36.04 bc | 42.83 b | 7.40 bc |
| | | S3 | 496.05 abc | 35.74 c | 40.94 b | 7.07 d | 494.45 bc | 35.18 c | 42.03 b | 7.22 d |
| | | S4 | 502.88 ab | 34.47 d | 40.05 cd | 6.68 ef | 507.80 abc | 34.03 d | 40.96 c | 6.90 e |
| | | S5 | 507.30 a | 33.25 e | 39.42 de | 6.54 g | 514.63 a | 32.74 e | 40.76 cde | 6.55 f |
| | | S6 | 506.73 a | 33.03 e | 39.16 ef | 6.28 h | 510.00 ab | 32.14 e | 39.94 ef | 6.33 g |

Data followed by different lower-case letters are significantly different at the 5% probability level as determined by the LSD test.

**Table 6.** Correlation between experimental point, sowing date and yield components (F value).

| | 2017–2018 | | | | 2018–2019 | | | |
|---|---|---|---|---|---|---|---|---|
| | Panicles | Grain per Panicle | Grain Weight | Yield | Panicles | Grain per Panicle | Grain Weight | Yield |
| latitude (L) | 5.35 * | 12.91 ** | 87.59 ** | 102.88 ** | 6.83 * | 14.34 ** | 89.29 ** | 101.24 ** |
| Variety (V) | 8.79 ** | 6.91 * | 0.18 | 3.92 | 4.06 | 19.44 ** | 3.97 | 0 |
| Sowing (S) | 10.76 ** | 259.65 ** | 171.57 ** | 716.10 ** | 6.19 ** | 234.29 ** | 92.71 ** | 441.29 ** |
| L × V | 1.39 | 11.70 ** | 0.65 | 1.04 | 2.81 | 2.12 | 6.15 * | 0.35 |
| L × S | 0.13 | 1.41 | 2.44 | 2.16 | 0.42 | 0.67 | 1.41 | 3.75 * |
| V × S | 0.77 | 1.1 | 0.35 | 0.56 | 0.65 | 2.42 | 1.86 | 0.27 |
| P × V × S | 0.34 | 0.82 | 1.05 | 0.59 | 0.4 | 0.65 | 0.68 | 0.65 |

Values are means ± SE. * and ** are significant at $p = 0.05$ and $p = 0.01$ levels, respectively.

### 3.4. Differences in Dry Matter Formation under Different Temperature and Radiation Conditions

The dry matter accumulation of the two spring varieties showed the same change pattern at different latitudes and when sowed on different sowing dates, and the annual change was also the same pattern (Table 7). When planted on the same date, the dry matter accumulation at HL was 0.12–0.91 t·ha⁻¹ less than LL. The difference in latitudes became increasingly smaller with the postponement of the sowing date. The dry matter accumulation during the whole growth period decreased gradually with the postponement of the sowing date. Compared with S1, the dry matter accumulation of S2–S6 decreased by 0.28–0.51·ha⁻¹, 0.7–0.91 t·ha⁻¹, 1.06–1.38 t·ha⁻¹, 1.41–2.08 t·ha⁻¹ and 1.68–2.38 t·ha⁻¹, respectively.

**Table 7.** Effects of different temperature and radiation on dry matter accumulation.

| Varieties | Latitude | Sowing | 2017–2018 | | | 2017–2018 | | |
|---|---|---|---|---|---|---|---|---|
| | | | JO | HE | MA | JO | HE | MA |
| YM23 | HL | S1 | 4.14 a | 9.64 a | 15.05 a | 4.24 a | 9.94 a | 15.24 a |
| | | S2 | 4.05 ab | 9.49 b | 14.76 b | 4.11 b | 9.73 b | 14.93 a |
| | | S3 | 3.77 c | 9.10 c | 14.13 c | 3.98 c | 9.51 c | 14.55 b |
| | | S4 | 3.64 d | 8.86 d | 13.73 d | 3.84 d | 9.25 d | 14.10 c |
| | | S5 | 3.50 e | 8.66 e | 13.30 e | 3.74 e | 9.10 e | 13.84 cd |
| | | S6 | 3.47 e | 8.61 e | 13.28 e | 3.70 e | 8.95 e | 13.56 d |
| | LL | S1 | 4.43 a | 10.33 a | 15.68 a | 4.64 a | 10.62 a | 16.07 a |
| | | S2 | 4.24 b | 10.06 b | 15.18 b | 4.47 b | 10.40 ab | 15.73 b |
| | | S3 | 4.14 bc | 9.82 c | 14.77 c | 4.22 c | 10.09 c | 15.23 c |
| | | S4 | 3.92 d | 9.61 d | 14.38 d | 3.99 d | 9.80 d | 14.74 d |
| | | S5 | 3.86 e | 9.40 e | 14.06 e | 3.65 e | 9.32 e | 13.99 e |
| | | S6 | 3.78 e | 9.23 f | 13.75 f | 3.58 e | 9.13 e | 13.69 e |

**Table 7.** *Cont.*

| Varieties | Latitude | Sowing | 2017–2018 | | | 2017–2018 | | |
|---|---|---|---|---|---|---|---|---|
| | | | **JO** | **HE** | **MA** | **JO** | **HE** | **MA** |
| YM25 | HL | S1 | 4.21 a | 9.85 a | 15.27 a | 4.33 a | 10.16 a | 15.52 a |
| | | S2 | 4.07 b | 9.71 b | 14.98 b | 4.22 a | 10.01 b | 15.22 b |
| | | S3 | 3.83 c | 9.45 c | 14.51 c | 4.07 b | 9.70 c | 14.71 c |
| | | S4 | 3.63 d | 9.26 d | 14.20 d | 3.88 c | 9.42 d | 14.23 d |
| | | S5 | 3.50 ed | 9.04 e | 13.74 e | 3.72 d | 9.16 e | 13.82 e |
| | | S6 | 3.33 e | 8.86 f | 13.42 f | 3.62 d | 9.02 f | 13.60 e |
| | LL | S1 | 4.49 a | 10.60 a | 15.94 a | 4.79 a | 10.81 a | 16.32 a |
| | | S2 | 4.34 b | 10.27 b | 15.43 b | 4.59 b | 10.56 b | 15.88 b |
| | | S3 | 4.20 c | 10.06 c | 15.10 bc | 4.53 b | 10.40 b | 15.62 b |
| | | S4 | 4.04 d | 9.81 d | 14.55 c | 4.25 c | 10.06 c | 15.08 c |
| | | S5 | 3.91 e | 9.59 e | 14.30 cd | 4.03 d | 9.73 d | 14.54 d |
| | | S6 | 3.82 e | 9.38 f | 13.94 d | 3.87 e | 9.43 e | 14.09 e |

Data followed by different lower-case letters are significantly different at the 5% probability level as determined by the LSD test.

### 3.5. Correlations between Temperature, Radiation and Yield at Different Latitudes

There was a very significant positive correlation between the effective temperature accumulation and yield in the whole growth stage and a very significant negative correlation between mean daily temperature and yield in the whole growth period, but the differences in the mean daily temperature in different years and at different latitudes were quite pronounced (Figure 1). The accumulated solar radiation and yield in the whole growth period had a significant positive correlation and a very significant negative correlation between the mean daily solar radiation and yield in the whole growth period (Figure 2). The yield/effective accumulated temperature reflects the temperature productivity (Figure 3). The temperature productivity at HL was higher than LL. The temperature productivity at HL decreased gradually, followed by S2 and at the lower latitude it gradually decreased, followed by S3. The yield/accumulated solar radiation represents the radiation productivity (Figure 4). The radiation productivity at HL was lower than LL, and the radiation productivity at both latitudes reached the maximum at S1 or S2.

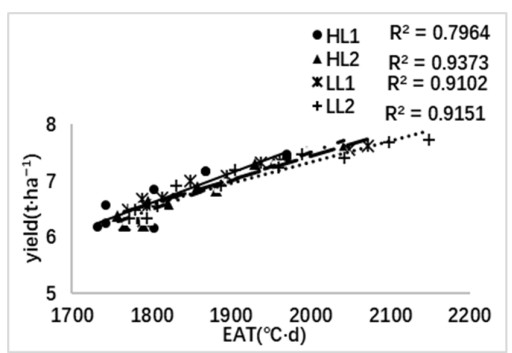 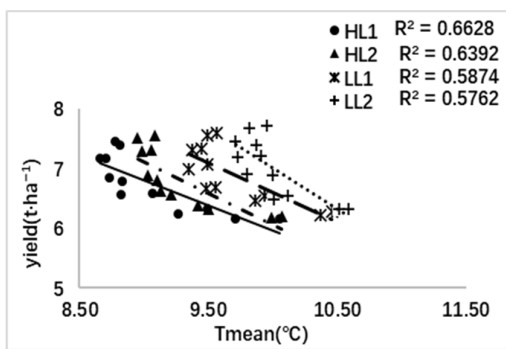

**Figure 1.** Correlation between temperature and yield during the whole growth period. HL1: 2017–2018 Donghai County test site, HL2: 2018–2019 Donghai County test site, LL1: 2017–2018 Jianhu County test site, LL2: 2018–2019 Jianhu County test site.

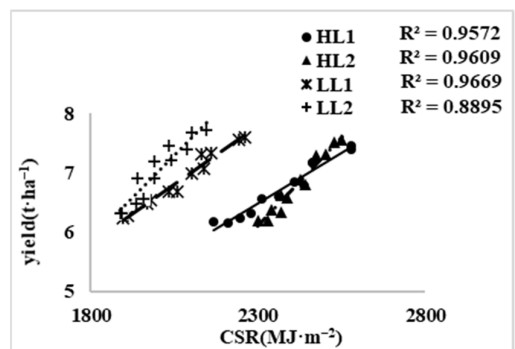 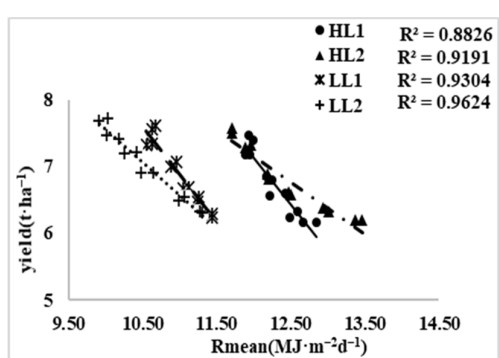

**Figure 2.** Correlation between radiation and yield during the whole growth period. HL1: 2017–2018 Donghai County test site, HL2: 2018–2019 Donghai County test site, LL1: 2017–2018 Jianhu County test site, LL2: 2018–2019 Jianhu County test site.

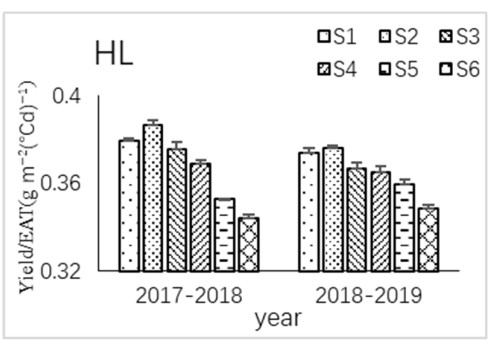 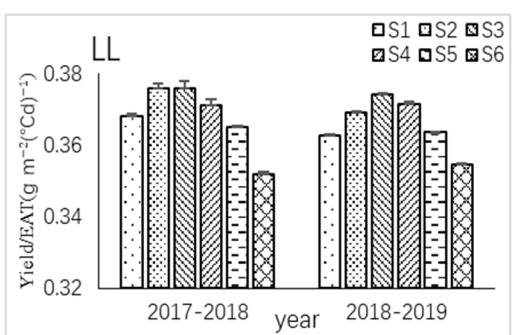

**Figure 3.** The ratio of yield to effective accumulative temperature at two latitudes in 2017–2018 and 2018–2019. S1, S2, S3, S4, S5 and S6 represent the sowing dates of 31October, 10 November, 20 November, 30 November, 10 December and 20 December, respectively. HL: Donghai County test site, LL: Jianhu County test site. The vertical strip represents the standard error of the average value of the two varieties (nasty 2).

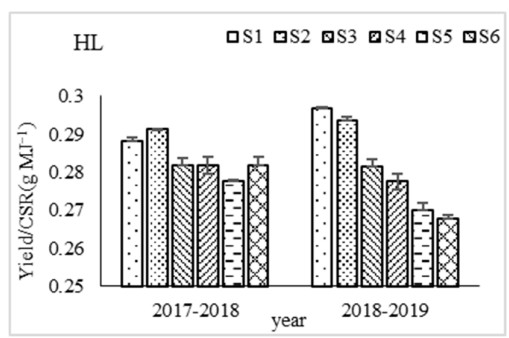 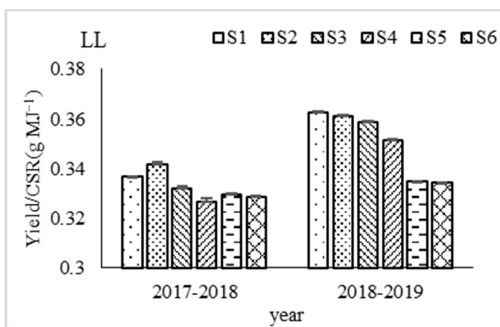

**Figure 4.** The ratio of yield to cumulative solar radiation at two latitudes in 2017–2018 and 2018–2019. S1, S2, S3, S4, S5 and S6 represent the sowing dates of31 October, 10 November, 20 November, 30 November, 10 December and 20 December, respectively. HL: Donghai County test site, LL: Jianhu County test site. The vertical strip represents the standard error of the average value of the two varieties (nasty 2).

### 3.6. Correlations between Temperature, Radiation and Dry Matter Accumulation at Different Latitudes

The correlations between dry matter accumulation and temperature were the same at different latitudes (Tables 8 and 9). In the whole growth period, from seedling emergence to jointing stage, jointing to the heading stage and heading to the maturity stage, there was a very significant positive correlation between the effective accumulated temperature and

dry matter accumulation, while the correlation between mean daily temperature and dry matter accumulation was significantly negative. There was a significant positive correlation between accumulated solar radiation and dry matter accumulation in the whole growth period and the each growth period. There was a significant negative correlation between the mean daily solar radiation and dry matter accumulation in the jointing to heading stage and the heading to maturity stage at Jianhu County.

**Table 8.** Correlation between temperature and radiation on dry matter of spring wheat in different periods at Donghai County.

|  | EM-JO | JO-HE | HE-MA | Whole Growth Period |
|---|---|---|---|---|
| EAT | 0.9501 ** | 0.6463 ** | 0.9745 ** | 0.9451 ** |
| $T_{mean}$ | −0.9444 ** | −0.6159 * | −0.9579 ** | −0.7970 ** |
| CSR | 0.9666 ** | 0.6324 * | 0.9823 ** | 0.9862 ** |
| $R_{mean}$ | −0.9630 ** | −0.3131 | 0.3365 | −0.9445 ** |

Values are means ± SE. * and ** are significant at $p = 0.05$ and $p = 0.01$ levels, respectively.

**Table 9.** Correlation between temperature and radiation on dry matter of spring wheat in different periods at Jianhu County.

|  | EM-JO | JO-HE | HE-MA | Whole Growth Period |
|---|---|---|---|---|
| EAT | 0.9762 ** | 0.8576 ** | 0.9712 ** | 0.9063 ** |
| $T_{mean}$ | −0.7918 ** | −0.7183 ** | −0.9288 ** | −0.7363 ** |
| CSR | 0.9156 ** | 0.8644 ** | 0.8702 ** | 0.9899 ** |
| $R_{mean}$ | −0.8977 ** | 0.5891 * | −0.7970 ** | −0.9390 ** |

Values are means ± SE. * and ** are significant at $p = 0.05$ and $p = 0.01$ levels, respectively.

## 4. Discussion

### 4.1. Effects of Latitude and Sowing Date on Temperature and Radiation Received by Wheat

Temperature and radiation are the main climatic factors affecting wheat yield [20,21]. Previous studies tried to improve the temperature and radiation utilization efficiency by changing the planting methods or adjusting the cultivation measures [22,23]. In this study, the effects of temperature and radiation on spring wheat were studied by choosing different sowing dates for different latitudes. For each latitude change at the ecological points, the number of growing days of wheat decreased or increased by 3 to 4 days, and the average temperature of the whole growing period increased or decreased by 0.2 °C [24]. Studies have shown that with an increase in the latitude, the mean daily temperature decreases, the effective accumulated temperature decreases, and the radiation increases in the whole growth period [25,26]. This study showed that with an increase in the planting latitude, the maturity stage of spring wheat was delayed by 4–5 d, the average temperature decreased by 0.66 °C, the effective accumulated temperature decreased by 50 °C d on average, and the accumulated solar radiation increased by 360 MJ m$^{-2}$ on average. The delayed sowing date led to the delay of maturity, the shortening of the growth stage, an increase in mean daily temperature, and decrease in the effective accumulated temperature and radiation [27,28]. For each postponement of the sowing date in the study, the mean daily temperature increased by 0.18 °C, the effective accumulated temperature decreased by 53 °C d, and the accumulated solar radiation decreased by 60 MJm$^{-2}$.

### 4.2. Effects of Temperature and Radiation on Wheat Material Production

Temperature and radiation have important effects on the formation of wheat matter. Increasing the accumulated temperature and accumulated solar radiation can improve dry matter accumulation and make them easy to transport [29,30]. When accumulated temperature reaches a certain value, the effect of increased accumulated temperature on biomass accumulation will decrease. An increase in mean daily temperature was

not conducive to biomass accumulation [31]. In this study, the mean daily temperature was negatively correlated with dry matter accumulation, while accumulated temperature and accumulated solar radiation were positively correlated with dry matter accumulation. Interestingly, there was no correlation between daily radiation and dry matter accumulation in the jointing to heading stage and in the heading to maturity stage at a high latitude. This may be due to the mixed effect of temperature and radiation, which was that too high a temperature will have a negative impact on the plants, and higher radiation will not be fully utilized [32].

An increase in dry matter accumulation contributes to the relatively high yield [33]. In this study, the performance of yield was similar to the dry matter accumulation, for which a decrease was observed as accumulated temperature and accumulated solar radiation decreased. The gradual decrease in the yield was mainly due to the decrease in the grain number per spike and 1000-grain weight. At the suitable spike differentiation temperature, a high temperature accelerated spikelet differentiation and reduced spikelet floret [34,35]. A high temperature accelerates grain filling, shortens grain filling time and decreases 1000-grain weight [11,36]. In this study, when the sowing date for the jointing to heading stage was delayed, the temperature increased by 0.5 °C and the accumulated temperature decreased by 7.71 °C d. In the heading to maturity stage, the mean daily temperature increased by 0.32 °C and accumulated temperature decreased by 14.96 °C d when the sowing date was delayed. Therefore, the decrease in the effective accumulated temperature and accumulated solar radiation and an increase in mean daily temperature led to a decrease in the grain number per spike and 1000-grain weight, and finally led to the decreased yield.

### 4.3. Utilization Efficiency of Temperature and Radiation at Different Latitudes

The temperature and radiation productivity of wheat gradually decreased with a decrease in latitude and the postponement of the sowing date [37,38]. In this study, the temperature productivity at Donghai County was higher than Jianhu County and the radiation productivity at Donghai County was lower than Jianhu County. The influence of temperature led to the unfulfilled use of higher radiation resources. At Donghai County, temperature and radiation productivity decreased gradually after S2. At Jianhu County, the productivity decreased gradually after S3, and the radiation productivity decreased gradually after S2. Therefore, with the postponement of the sowing date, the productivity of temperature and radiation increased at first and then decreased.

### 4.4. Adaptability of Growing Spring Wheat Northward

In the north of Jiangsu province, the stubble problem under the Rice-Wheat Rotation system is more serious, and requires adjustments. Global warming has provided the possibility for the spring wheat to move northward by one latitude and to grow in those regions [39]. There is little difference in the length of sunshine between the two places, and the wheat can blossom and produce grain at the right time. Before moving the spring wheat growth region northward, overwintering temperature and cold spells in late spring have to be considered. One issue is the frost injury that may occur in the tillering stage and another is the late spring cold at the jointing stage and booting stage. Generally speaking, the spring varieties can withstand a low temperature of −10 °C [40]. In this study, neither of these two experimental sites had a temperature as low as −10 °C, and all wheat crops were unaffected by cold spells in late spring. Negative accumulated temperature is an index with which to investigate the frost injury that may be inflicted upon wheat growth in winter and to assess whether wheat can be introduced or not [41]. In this study, the negative accumulated temperature in the past two years trended towards a higher position between the middle and high levels, and the lowest negative accumulated temperature at HL was as low as −74.9 °C. Only a small portion of leaves of YM23 in S1 yellowed, and other sowing stages were not affected by low temperature. Wheat is subjected to high temperature stress when the temperature exceeds 25 °C in the filling stage [11,42]. During the experiment in this study, the temperature at HL was over 25 °C for about 7–8 days, and

the temperature difference between the latitudes was small. The influence of hot and dry air on wheat growth was less pronounced. Under the unified management of diseases and insect pests in the field, no serious diseases and insect pests appeared in wheat, and the incidence of powdery mildew in wheat planted in the north was less than that in wheat planted in the LL. If the spring varieties growth region moves northward, their yield would decrease, but the wheat yield from all sowing stages was higher than the average yield of wheat in the northern Jiangsu region (6 t ha$^{-1}$). Compared with semi-winter varieties, spring varieties have a shorter growth time and are not as demanding in terms of cold temperatures during vernalization as the semi-winter varieties [43]. In this experiment, moving the spring wheat variety northward by one latitude could alleviate the stubble problem and recover some yield loss. However, there is still some risk of spring wheat moving northward, and it is necessary to screen spring wheat varieties with early maturity and strong resistance due to overwintering and cold spells in later spring.

## 5. Conclusions

In this study, the effects of temperature and radiation on the yield of spring wheat were investigated by setting different sowing dates for different latitudes. If spring wheat moved northward, it would result in a decrease in the effective accumulated temperature and mean daily temperature, an increase in accumulated solar radiation, and decrease in yield and dry matter accumulation. The postponement of sowing date leads to the postponement of the growth stage, a reduction in growth days, a decrease in the effective accumulated temperature and accumulated solar radiation, and an increase in daily temperature. The mean daily temperature increased and the effective accumulated temperature and accumulated solar radiation decreased, which resulted in a reduction in the grain number per spike and 1000-grain weight. The effective accumulated temperature and accumulated solar radiation had significant positive correlations with yield, while the mean daily temperature had significant negative correlations with yield. There was a significant positive correlation between the effective accumulated temperature and accumulated solar radiation and dry matter accumulation, while there was a significant negative correlation between daily temperature and dry matter accumulation.

**Author Contributions:** Conceptualization, H.W.; methodology, N.Z.; validation, H.Z. and H.G.; formal analysis, Z.Z.; investigation, Z.Z. and B.L.; resources, Z.Z. and J.T.; writing—original draft preparation, Z.Z.; writing—review and editing, Z.Z. and Z.X.; supervision, H.Z. and H.W.; project administration, H.Z. All authors have read and agreed to the published version of the manuscript.

**Funding:** Hongcheng Zhang, Jiangsu Demonstration Project of Modern Agricultural Machinery Equipment and Technology, Yangzhou University: (NJ2020-58, NJ2019-33, NJ2021-63).

**Institutional Review Board Statement:** Not applicable.

**Informed Consent Statement:** Not applicable.

**Data Availability Statement:** Not applicable.

**Conflicts of Interest:** The authors declare no conflict of interest.

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
