# Peer review of "Effects of Temperature and Radiation on Yield of Spring Wheat at Different Latitudes"

_agriculture, doi:10.3390/agriculture12050627_

Round 1

Reviewer 1 Report

This article is an interesting one with important findings documented. However, there are some concerns as mentioned below:

  1. In the Introduction part, there is no mention of proper research-gap. As this is a well known fact and many researches is going on globally. Authors must describe how their work is innovative. What are the hypothesis? Where is the novelty statement? All these must be included in the Introduction part.
  2. In the methodology part, the experimental data collection as well as statistical design must be mentioned clearly. Both of these parts are incomplete in this manuscript.
  3. Results part is well written with sufficient illustration of the data obtained in this study.
  4. Line no. 267-269: It is a controversial statement as in case of wheat increased temperature not always increase the biomass production, rather, terminal heat-stress results in reduction in biomass production.
  5. In section 4.4: the adaptability part needs a complete revision considering all the possible concerns on wheat production.   

Reviewer 2 Report

  • The authors of the article repeatedly use the imprecise term "low / hight latitude" to define the location of the fields on which the experiments were carried out. Unfortunately, the content of the publication does not indicate whether the height of the land for the analyzed areas was comparable.
  • The authors state that the experiment covered the area where the cultivation of the "Rice-Wheat Rotation system" was used. They report the yields obtained earlier for rice. However, it is not known whether wheat was grown there before the experiment, and if so, what its crops were.
  • What was the guiding principle behind the selection of certain types/varieties of wheat?
  • What was the guiding principle behind the date of sowing wheat assumed in the experiment?
  • From what was the fertilizer doses (line 91) determined?
  • On what specific dates did the successive phases of the growth of the thorns occur for the variants of the experiment?
  • The authors only write about meteorological data on air temperature and sunshine hours obtained from weather stations (lines 107-108). How was solar radiation determined?
  • It is not true that only "Temperature and radiation are the main environmental factors affecting wheat yield" (line 248). Certainly, they also include soil and water conditions.
  • Why did the authors not compare the water conditions of wheat cultivation (e.g. the amount of precipitation)?
  • The authors cite global climate change as a determinant of changes in plant cultivation. This is a recent common occurrence. Unfortunately, in relation to this publication, it is difficult to assess the obtained results presented in this context.
  • Some technical notes: 1) (line 42) citation of "AssengS"; 2) (line 82) "kg-1"; 3) (line 97) consistently in the text, "sowing date" should be described symbolically as "SO"; 4) the title of table 2, ".... daily temperature of wheat ..." (line 153) requires correction;

Round 2

Reviewer 1 Report

The paper has been modified as recommended. It can be accepted for publication now.